# Lateral Rectus Muscle Resection for New-Onset Esotropia Following Medial Orbital Wall Decompression in Thyroid Eye Disease

**DOI:** 10.3390/medicina61040559

**Published:** 2025-03-21

**Authors:** Jonnah Kristina Teope, Naomi Umezawa, Yasuhiro Takahashi

**Affiliations:** 1Department of Oculoplastic, Orbital and Lacrimal Surgery, Aichi Medical University Hospital, Nagakute 480-1195, Aichi, Japan; jkristina.teope@gmail.com (J.K.T.); nume0828@yahoo.co.jp (N.U.); 2Department of Ophthalmology, Aichi Medical University, Nagakute 480-1195, Aichi, Japan

**Keywords:** lateral rectus muscle resection, esotropia, medial orbital wall decompression, thyroid eye disease, dysthyroid optic neuropathy

## Abstract

*Background and Objectives:* The purpose of this study was to evaluate the outcome of lateral rectus (LR) muscle resection for new-onset or worsening esotropia after medial orbital wall decompression in patients with thyroid eye disease. *Materials and Methods:* This retrospective observational study included 20 patients. Preoperative and postoperative measurements of ocular deviation angles and fields of binocular single vision (BSV) were performed one day before and three months after surgery. Surgical success was defined as postoperative horizontal ocular deviation ≤ 5° and BSV including the primary position. Factors influencing a reduction in the esodeviation angle were analyzed using univariate and multivariate linear regression analyses. *Results:* Eighteen patients (90.0%) were deemed as successful surgical cases. The esodeviation angle decreased from 19.4 ± 11.2° to 1.0 ± 2.6°. In multivariate analysis, a reduction in the esodeviation angle was correlated with the presence of dysthyroid optic neuropathy (*p* = 0.027), amounts of LR muscle resection in mild eyes (*p* = 0.014), and amounts of additional medial rectus muscle recession in severe eyes (*p* < 0.001). *Conclusions:* LR muscle resection showed a high success rate in correcting new-onset or worsening esotropia which developed after medial orbital wall decompression. Several factors influencing a reduction in the esodeviation angle were found.

## 1. Introduction

New-onset or worsening esotropia after medial orbital wall decompression is common in patients with thyroid eye disease (TED) [1,2,3,4,5,6,7]. Incidences of new-onset diplopia after medial orbital wall decompression, balanced decompression (medial + deep lateral orbital wall decompression), and inferomedial orbital wall decompression are 10–35%, 33%, and 45–75%, respectively [3]. This complication negatively affects patients’ quality of life.

Esotropic shift post-decompression can develop because of anatomic displacement of the medial rectus (MR) muscle into the ethmoid sinus [2,8,9,10,11,12]. In addition, patients with an enlarged MR muscle have a high risk of post-decompression esotropia [9,11]. An elongated, medially shifted MR muscle excessively pulls the globe toward the medial side, leading to esotropia and limited abduction [8,11], although the maximum active force of an elongated MR muscle is reduced [12].

MR muscle recession has been the preferred surgical approach to correct esotropia in TED, as it is more conservative than resection [13]. Resection is generally avoided due to risk of worsening inflammation, sequential exotropia, and convergence difficulties [14]. However, we assumed that lateral rectus (LR) muscle resection is physiologically eligible for the correction of post-decompression esotropia, because a resected LR muscle has increased tractional force, rectifies the horizontal muscle imbalance, and can pull back the medially tugged globe. MR muscle recession further reduces the maximum active force of an elongated, medially shifted MR muscle.

Success rates of strabismus surgery are comparable between patients who have undergone orbital decompression and those who have not [15,16,17]. However, patients who have had decompression often require combined MR muscle recession and LR muscle resection, as MR muscle recession alone frequently leads to undercorrection [1,18,19,20]. LR muscle resection as a primary intervention in restoring ocular alignment and binocular single vision (BSV) for post-decompression esotropia has yet to be discussed. Our study evaluated the outcome of LR muscle resection for new-onset or worsening esotropia after bony orbital decompression, including the medial orbital wall decompression, in TED.

## 2. Materials and Methods

### 2.1. Study Design

This was a retrospective, observational study, including patients with TED who underwent LR muscle resection for esotropia which developed or worsened after medial orbital wall decompression or 2-wall orbital decompression including the medial orbital wall, from January 2015 to December 2023. Orbital wall decompression and strabismus surgery were performed by one of the authors (YT), except medial or inferomedial orbital wall decompressions via the endonasal endoscopic approach, which were performed by otorhinolaryngologists in patients with dysthyroid optic neuropathy (DON) for prevention of intraoperative vision loss [21]. Decimal visual acuity recovered to at least 0.5 after orbital decompression in all of the patients with DON. Gaze restriction was graded on the Gorman score from 0 (normal) to 3 (severely restricted), and eyes with larger scores were determined as severe eyes [22]. Patients were excluded from this study if they had insufficient clinical data, a history of strabismus surgery prior to LR muscle resection, a follow-up period of less than 3 months, concurrent neuro-ophthalmologic disorders, or an intracranial lesion.

### 2.2. Diagnosis of TED

A diagnosis of TED was established based on Bartley’s criteria [23]. DON was diagnosed based on optic nerve compression caused by enlarged extraocular muscles (EOMs), as confirmed by imaging, along with one or more of the following: best corrected decimal visual acuity of 0.5 or worse, visual field defects, papilledema or optic disk pallor, reduced color vision, and a positive relative afferent pupillary defect [21]. All patients included in the study were controlled as euthyroid at the time of strabismus surgery, and were in the static or chronic “burnout” phase of TED, confirmed based on the absence of inflammation in the EOMs on T2-weighted fat-suppressed magnetic resonance images [24,25].

### 2.3. Data Collection

The following data were collected: sex, age, smoking history, history of steroid pulse therapy and/or orbital radiotherapy, presence or absence of DON, procedures of preceding orbital decompression surgery, amounts of orbital fat removed during orbital decompression, Hertel exophthalmometric values measured one day before and three months after orbital decompression, procedures of strabismus surgery, amounts of EOM resection, recession, and transposition, and ocular deviation angles and fields of BSV measured one day before and three months after strabismus surgery. The smoking status was classified, based on the number of cigarettes smoked per day, as follows: 0, no smoking; 1, ˂10 cigarettes/day; 2, 10–20 cigarettes/day; and 3, ˃20 cigarettes/day [26]. Patients who had smoked previously but stopped ≥ 2 years prior to examinations were considered non-smokers [27]. The dose of orbital radiotherapy administered was 20 Gy in all treated patients [21].

Since alternate prism and cover tests were not performed in a few patients, the horizontal ocular deviation angle was measured on a Hess chart using ImageJ software version 1.54g (National Institute of Health, Bethesda, MD, USA), based on a previous report (Figure 1A) [9]. Inter-observer reliability was not required, as all measurements were made using the validated software. The horizontal length of one scale (5° per small box) was initially measured, followed by the distance from the central vertical line to the center of the resultant Hess chart. The angle of ocular deviation in the horizontal direction was then calculated using this formula: (5 × the horizontal length from the central vertical line to the resultant Hess chart)/the horizontal length of 1 scale. When the central dot on Hess chart was not identified due to severe esotropia, the dot corresponding to the lateral rectus muscle motility was used to measure the deviation angle. Vertical and torsional ocular deviation angles were measured using a synoptophore, because proximal convergence affects the measurement results of the horizontal deviation angle obtained using a synoptophore, and the torsional deviation angle cannot be measured using a Hess chart. Two slides were employed: one using a black circle with a cross-shaped blank, and the other using a black cross (L-25G; Inami, Tokyo, Japan). The deviation angles were measured in the primary position, as well as at 15° upward and 15° downward gaze.

The areas of fields of BSV were also measured using ImageJ software, based on a previous report [9]. We then determined the normal range of the BSV field in a Japanese population, based on earlier research [28]. Next, the pre- and postoperative areas of the BSV fields were measured (Figure 1B). The percentage of the pre- and postoperative BSV fields relative to the normal BSV field (%BSV) was calculated. The BSV field results were classified into five categories (B1 to B5): B1, within the normal range (±2 × standard deviations); B2, the BSV field extends by at least 20 degrees superiorly, 40 degrees inferiorly, and 30 degrees horizontally; B3, a smaller BSV field than B2, but still includes the primary gaze; B4, the BSV field excludes the primary gaze; and B5, no measurable BSV field in any direction of gaze [28].

### 2.4. Computed Tomography (CT)

Contiguous 1 mm axial and coronal CT images (Aquilion Precision, Canon, Tokyo, Japan, or SOMATOM AS plus, Siemens Japan K.K., Tokyo, Japan) were obtained using the soft tissue window algorithm, prior to and one day post-orbital decompression The maximum cross-sectional areas of the LR and MR muscles were assessed on both pre- and post-decompression CT images [29]. The LR muscle area was measured on the axial CT images that displayed the entire length of the muscle (Figure 1C). Coronal CT does not depict the exact measurement of the LR muscle in the anteroposterior direction, due to the muscle’s oblique orientation [29]. The maximum cross-section of the MR muscle was measured on the coronal CT showing the maximum cross-section of the MR muscle (Figure 1D). All measurements were performed using the image viewing software (ShadeQuest/ViewR; Yokogawa Medical Solutions Corporation, Tokyo, Japan) by one of the authors (YT).

The EOMs expand after orbital decompression [29]. The increasing rate of the cross-sectional areas of the LR and MR muscles were calculated as follows: [(post-decompression area − pre-decompression area)/pre-decompression area] × 100.

### 2.5. Procedures for Strabismus

Strabismus surgery was performed under general anesthesia. A forced duction test was performed to confirm restriction in both the medial and lateral directions. A perilimbal conjunctival incision with radial relaxing incisions was made in the temporal quadrant to expose the LR muscle for resection. The LR muscle was secured at its insertion with a muscle hook, and was dissected from the Tenon’s capsule with cotton swabs. A caliper was used to measure the width of the LR muscle tendon at the scleral insertion. Locking 6-0 or 8-0 polyglactin sutures (Vicryl^®^; Johnson and Johnson Company, New Brunswick, NJ, USA) were placed at the superior and inferior margins of the LR muscle, posterior to the point approximated based on the preoperative esodeviation measurement. The amount of LR resection was calculated using the following formula: 1° of esodeviation per 1 mm of LR resection. A 2–3° residual esotropia was targeted to account for late exotropic drift. The LR muscle was then detached from its insertion. It was repositioned at the LR muscle insertion using the previously placed sutures, with additional fixation at 2 to 4 points to prevent muscle slippage. When correcting cyclotropia, the LR muscle was transposed vertically along the spiral of Tillaux. The amount of vertical LR muscle transposition was preoperatively calculated based on the torsional angle and tendon width measurements, with 8° of cyclotropic correction achieved per 1 tendon width of LR muscle transposition [30]. The conjunctiva was closed using 8-0 polyglactin sutures. Superior rectus muscle resection was performed in a similar manner.

In patients with combined recession of the MR or inferior rectus muscle, the EOM tendon was secured at two points, 1 mm posterior to the globe insertion, to account for the 1 mm thickness of the muscle hook. The muscle was then detached from its insertion. Sutures were placed on the sclera 1 mm posterior to the estimated location based on the preoperative ocular deviation angle. The amount of recession was calculated using the following formula: 2° of deviation angle per 1 mm of muscle recession [30]. Additional fixation of the muscle tendon to the sclera was performed at 2 to 4 points.

After surgery, 125 mg of intravenous methylprednisolone for 3 days, an oral antibiotic for 3 days, and a topical antibiotic and steroid were administered.

Surgical success was defined when patients fulfilled the following 2 conditions: (1) a postoperative angle of horizontal ocular deviation of ≤5°; and (2) a postoperative BSV grade of B3 or better. The surgical success criteria were based on prior research and clinical standards [22,27,30,31,32].

### 2.6. Statistical Analyses

Patient data and measurement results were presented as means ± standard deviations. The preoperative value was subtracted from the postoperative value to determine the postoperative changes in measurement results. The horizontal deviation angle measured in a 15° downward gaze was subtracted from that measured in a 15° upward gaze to assess the strabismus pattern. With an approximation of 2 prism diopters equating to 1°, subtraction values of 5° or more were considered indicative of clinically significant A-pattern strabismus, and values of −7.5° or more were considered indicative of V-pattern strabismus [33]. Statistical comparisons of the preoperative and postoperative measurements were conducted using a paired t-test or chi-square test. Univariate analyses, followed by multivariate analyses with stepwise variable selection, were performed. All statistical analyses were conducted using SPSS™ version 26 software (IBM Japan, Tokyo, Japan). A two-tailed *p* value < 0.05 was considered statistically significant.

## 3. Results

Patient characteristics and pre-strabismus surgery conditions are shown in Table 1. Out of the total 314 patients who underwent orbital decompression, 30 patients underwent correction of strabismus which developed or worsened after orbital decompression, and only 20 patients met the inclusion criteria.

Details of the strabismus surgery are shown in Table 2. LR muscle resection was performed bilaterally in 17 patients (85.0%) and unilaterally (only in severe eyes) in 3 patients (15.0%). The amount of LR muscle resection was 7.0 ± 2.9 mm in severe eyes and 4.6 ± 3.4 mm in mild eyes. The LR muscle was vertically transposed in five patients for correction of cyclotropia. Six patients underwent additional strabismus surgery.

The outcomes of strabismus surgery are shown in Table 3. Eighteen patients (90.0%) were classified as having successful surgical outcomes, while the remaining two patients (10.0%) were classified as unsuccessful cases. Esotropia was undercorrected (postoperative changes in esodeviation angle, 10.0° to 5.4°) in one patient, although the postoperative field of BSV was B1. The other patient had a postoperative BSV of B4, due to changes in the vertical deviation angle (4° to −5°) after bilateral LR muscle resection and MR muscle recession, although esotropia was well corrected (postoperative esodeviation deviation angle of −2.1°) (Figure 2). None of the patients experienced overcorrection of esotropia (postoperative esodeviation angle of <−5°). Before surgery, the angles of esotropia, hypertropia, and excyclotropia were 19.4 ± 11.2°, 0.4 ± 2.3°, and 2.0 ± 3.0°, respectively. Postoperatively, the angles of esotropia, hypertropia, and excyclotropia were 1.0 ± 2.6° (*p* < 0.001), −0.4 ± 1.7° (*p* = 0.158), and 1.1 ± 1.4° (*p* = 0.095), respectively. The %BSV significantly increased from 28.1 ± 31.0% to 54.9 ± 28.3% (*p* = 0.005). One patient had a preoperative BSV grade of B1, but opted for surgical correction of deviation, due to diplopia during driving. The BSV grade improved after strabismus surgery (*p* = 0.005).

Table 4 shows the results of the regression analysis. Univariate analysis showed that there was correlation of postoperative changes in the esodeviation angle with the presence of DON (*p* = 0.016), the amounts of LR muscle resection in severe (*p* = 0.002) and mild eyes (*p* = 0.007), the amount of superior LR muscle transposition (*p* = 0.015), the amounts of MR muscle recession in severe (*p* < 0.001) and mild eyes (*p* = 0.002), and the amount of superior MR muscle transposition in mild eyes (*p* = 0.002). A change in the esodeviation angle was correlated with the presence of DON (*p* = 0.027), the amount of LR muscle resection in mild eyes (*p* = 0.014), and the amount of MR muscle recession in severe eyes (*p* < 0.001) (adjusted R^2^ = 0.728; *p* < 0.001), using multivariate stepwise analysis. The crude coefficients of each item were 7.279, 1.244, and 3.002, respectively. There was no multicollinearity among these factors.

## 4. Discussion

This study retrospectively examined the outcome of LR muscle resection for esotropia which developed after medial orbital wall decompression in TED. LR muscle resection had a 90% success rate in correcting new-onset esotropia following orbital wall decompression including the medial orbital wall. Our results were comparable to previous studies showing high rates of BSV restoration in TED after MR muscle recession, and even higher if deviation is not present prior to decompression [34,35]. Li et al. reported a 100% success rate of MR muscle recession for 11 cases of new-onset esotropia post decompression [34]. While two of our patients were deemed unsuccessful, our study had a relatively large population. One of our unsuccessful cases had a postoperative esodeviation just slightly above our surgical success criteria, and reported no diplopia post-surgery. The other patient was judged as unsuccessful due to a change in vertical deviation, but achieved successful correction of horizontal deviation.

New-onset or worsening esotropia after decompression has been highly attributed to postoperative centrifugal displacement of a weakened MR muscle [2,8,9,10,11]. MR muscle recession will further weaken the MR, while LR muscle resection can strengthen the LR enough to pull back a medially tugged globe. The absence of muscle fibrosis in such cases favors the safety of resection. However, the risk of resection aggravating inflammation and EOM restriction and overcorrection is high in patients with enlarged EOMs [19,20]. Other factors post decompression were then considered. Orbital decompression can increase MR muscle area and cause subsequent restriction of abduction [29]. Our results show that prior orbital decompression did not significantly affect the outcome of strabismus surgery, consistently with findings from previous studies [15,16]. The patients included in our study did not have TED reactivation, and were in the “burnout” phase at the time of strabismus surgery. Patients with DON or more severe TED tend to have more EOMs involved, causing not just esotropia, but other ocular misalignments as well, such as vertical and pattern deviations. Notably, all our patients obtained favorable esotropia correction without adverse outcomes, even after resection of an enlarged LR muscle. Similar outcomes were also demonstrated in previous studies [19,20,34]. However, our study was the first to focus on LR resection in patients with esotropia, specifically following medial orbital wall decompression.

In the present study, three patients (15.0%) had large-angle esotropia that required additional MR muscle recession. In the study presented by Li et al., seventy percent of patients who underwent orbital decompression underwent additional LR muscle resection after MR muscle recession [34]. Combining MR recession and LR resection has been advocated as primary treatment for large-angle esotropia before or after orbital decompression, as MR recession alone often results in undercorrection [36]. This combined surgical technique has resulted in a few cases of overcorrection that eventually resolved over time [36]. In our study, none of the patients had overcorrection. The amounts of LR muscle resection and MR muscle recession in both mild and severe eyes were correlated with postoperative changes in esodeviation angle in univariate analyses. However, only the amounts of LR muscle resection in mild eyes and MR muscle recession in severe eyes were identified as significant factors influencing a reduction in the esodeviation angle in multivariate analysis. The crude coefficients of each item were 1.244 and 3.002, respectively. Although there has been no study showing the dose-effect coefficient of EOM resection for restrictive strabismus in TED, the crude coefficient of MR muscle recession in severe eyes seems to be larger than previously reported results in mixed patients with and without a history of orbital decompression (3.44–3.93 prism) [37,38].

Eckstein et al. noted that MR muscle recession had a lower and more variable dose-effect after orbital decompression [2], and the study of Garrity et al. showed that LR muscle resection gave more predictable results than MR muscle recession for residual esotropia in TED [36]. However, multivariate analysis of our study demonstrated that the amounts of LR muscle resection in severe eyes and MR muscle recession in mild eyes do not significantly influence the reduction in the esodeviation angle. In cases where only LR muscle resection is planned, the postoperative reduction in esodeviation angle is estimated using the results of univariate analyses. The results of multivariate analysis indicate less predictability of combined LR muscle resection and MR muscle recession for large-angle esotropia which has developed after medial orbital wall decompression. It is important for both surgeons and patients to be aware of the potential unpredictability of outcomes in such cases. Surgeons should inform patients about the possibility of needing additional interventions or adjustments, and may want to explore alternative surgical approaches.

This study is limited by its retrospective design, relatively small number and heterogeneity of patients, and short follow-up period. A 3-month follow-up may not reveal late overcorrections and the possible effect of TED reactivation in the resected muscle. However, possible late exotropic drifts may spontaneously resolve over time due to fusional vergences [39]. We therefore anticipate the LR muscle resection performed in our study to yield stable long-term results. This is further supported by consistent outcomes observed with LR muscle resection for thyroid-associated esotropia, which have demonstrated stability after 19–23.5 months [34,36].

## 5. Conclusions

LR muscle resection showed a high success rate in correcting new-onset or worsening esotropia which developed after bony orbital decompression, including medial orbital wall decompression, in patients with TED. A postoperative reduction in esodeviation angle is predictable in cases where only LR muscle resection is performed. However, the surgical outcome of combined LR muscle resection and MR muscle recession is less predictable in cases with large-angle esotropia.

## Figures and Tables

**Figure 1 medicina-61-00559-f001:**
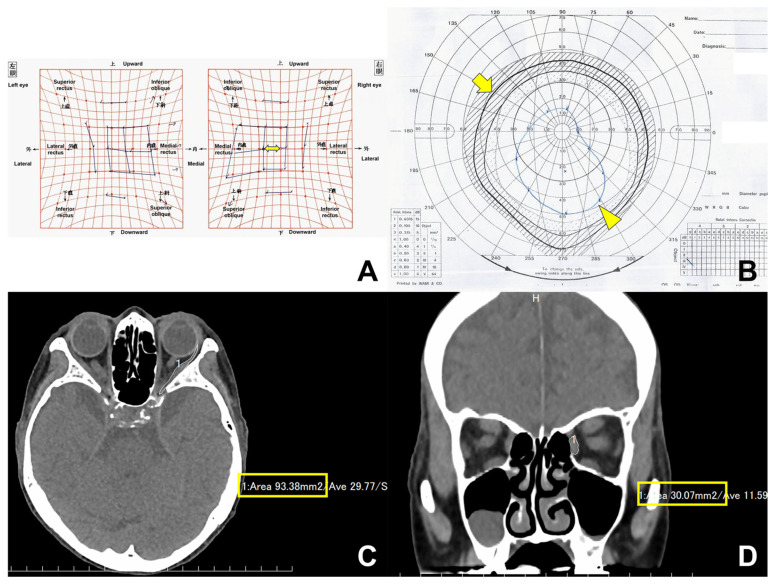
Measurements of (**A**) esodeviation angle (arrow) on Hess chart, (**B**) normal (arrow) and actual fields of binocular single vision (arrowhead), and (**C**,**D**) cross-sectional areas of lateral and medial rectus muscles on axial and coronal computed tomographic images.

**Figure 2 medicina-61-00559-f002:**
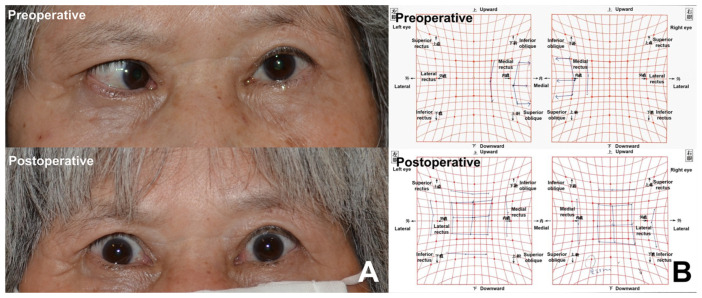
Face photos (**A**) and Hess chart measurements (**B**) of a patient with large-angle esotropia which developed after orbital decompression for dysthyroid optic neuropathy. Esotropia was well corrected after strabismus surgery.

**Table 1 medicina-61-00559-t001:** Data on patient characteristics and pre-strabismus surgery conditions.

	Total	R	L	Severe Eye	Mild Eye
Number of patients (M/L)	20 (2/18)				
Age (years)	54.0 ± 18.6				
Smoking status (0/1/2/3)	18/0/1/1				
Previous treatment history for TED					
Steroid pulse therapy	14				
Orbital radiation therapy	5				
Number of patients with DON (bilateral/unilateral)	8 (6/2)				
Orbital decompression techniques					
Bilateral balanced decompression	10				
Balanced decompression + medial decompression	2				
Bilateral inferomedial decompression	3				
Bilateral medial decompression	4				
Unilateral medial decompression	1				
Amount of removed orbital fat during balanced decompression (mL)		1.3 ± 1.1	1.3 ± 1.1		
Pre-decompression Hertel exophthalmometric value (mm)		20.3 ± 4.8	19.7 ± 4.9	20.2 ± 4.7	19.8 ± 5.0
Post-decompression Hertel exophthalmometric value (mm)		16.0 ± 4.4	15.1 ± 4.0	15.3 ± 4.1	15.8 ± 4.3
Proptosis reduction (mm)		4.3 ± 2.2	4.5 ± 2.2	4.8 ± 2.2	4.4 ± 2.2
*p* value		<0.001	<0.001	<0.001	<0.001
Cross-sectional area of LR muscle					
Pre-decompression (mm^2^)		154.0 ± 41.4	127.0 ± 34.6	139.2 ± 39.2	141.8 ± 41.6
Post-decompression (mm^2^)		171.4 ± 45.9	131.8 ± 33.5	154.0 ± 45.3	149.2 ± 44.7
Increasing rate (%)		13.2 ± 22.8	5.4 ± 18.6	11.4 ± 20.9	7.1 ± 21.3
*p* value		0.019	0.335	<0.001	<0.001
Cross-sectional area of MR muscle					
Pre-decompression (mm^2^)		55.1 ± 26.4	56.6 ± 28.8	59.2 ± 29.5	52.5 ± 25.2
Post-decompression (mm^2^)		65.5 ± 31.1	72.7 ± 38.6	71.3 ± 33.6	66.9 ± 36.6
Increasing rate (%)		20.4 ± 21.6	29.0 ± 39.9	22.9 ± 24.1	26.5 ± 38.9
*p* value		<0.001	0.004	<0.001	<0.001

M, male; F, female; R, right; L, left; TED, thyroid eye disease; DON, dysthyroid optic neuropathy; LR, lateral rectus; MR, medial rectus.

**Table 2 medicina-61-00559-t002:** Details of strabismus procedure.

	Total	Severe Eye	Mild Eye
Eyes with severe/mild EOM restriction			
R		8	12
L		12	8
LR resection (bilateral/unilateral)	20 (17/3)	20	17
Amount (mm)		7.0 ± 2.9	4.6 ± 3.4
LR vertical transposition (bilateral/unilateral)	5 (2/3)	3	4
Amount (tendon width) *		−0.3 ± 0.1	0 ± 0.2
Additional surgery			
Bilateral MR recession	1	1	1
Unilateral MR recession	2	2	0
(Additional superior MR transposition)	2	1	1
Unilateral SR resection	2	1	1
(Additional temporal SR transposition)	1	1	0
Bilateral IR recession	1	1	1
(Additional nasal IR transposition)	1	0	1

EOM, extraocular muscle; R, right; L, left; LR, lateral rectus; MR, medial rectus; SR, superior rectus; IR, inferior rectus. * Positive values indicate superior transposition.

**Table 3 medicina-61-00559-t003:** Comparison of surgical outcomes between successful and unsuccessful cases.

	Pre-Strabismus Surgery	Post-Strabismus Surgery	*p* Value
Ocular deviation angle (degrees)			
Esotropia	19.4 ± 11.2	1.0 ± 2.6	<0.001
Hypertropia	0.4 ± 2.3	−0.4 ± 1.7	0.158
Excyclotropia	2.0 ± 3.0	1.1 ± 1.4	0.095
Pattern strabismus			
A-pattern	1	0	
V-pattern	0	0	
%BSV (%)	28.1 ± 31.0	54.9 ± 28.3	0.005
BSV grade			
B1	1	3	0.005
B2	4	6	
B3	3	10	
B4	5	1	
B5	7	0	

BSV, binocular single vision.

**Table 4 medicina-61-00559-t004:** Results of univariate and multivariate analyses.

Predictive Factors	Univariate		Multivariate Stepwise	
	*p* Value	Crude Coefficient (95% CI)	*p* Value	Crude Coefficient (95% CI)
Previous history				
Steroid pulse therapy	0.122	9.035 (−2.646 to 20.716)		
Orbital radiation therapy	0.493	4.353 (−8.708 to 17.414)		
Smoking status	0.319	−3.547 (−10.817 to 3.722)		
Presence or absence of DON	0.016	12.587 (2.685 to 22.489)	0.027	7.279 (0.947 to 13.611)
Including deep lateral orbital wall decompression	0.402	−4.685 (−16.153 to 6.784)		
Including orbital floor decompression	0.126	11.478 (−3.535 to 26.491)		
Proptosis reduction				
Severe eyes	0.905	0.157 (−2.576 to 2.890)		
Mild eyes	0.829	−0.282 (−2.975 to 2.411)		
Increasing rate of cross-sectional area of LR				
Severe eyes	0.976	0.004 (−0.277 to 0.285)		
Mild eyes	0.808	−0.032 (−0.308 to 0.243)		
Increasing rate of cross-sectional area of MR				
Severe eyes	0.698	0.046 (−0.197 to 0.288)		
Mild eyes	0.561	0.042 (−0.108 to 0.192)		
Amount of LR resection				
Severe eyes	0.002	2.623 (1.096 to 4.150)	-	-
Mild eyes	0.007	2.074 (0.655 to 3.493)	0.014	1.244 (0.286 to 2.202)
Amount of superior LR transposition				
Severe eyes	0.015	−48.688 (−86.630 to −10.705)	-	-
Mild eyes	0.849	−2.959 (−35.068 to 29.149)		
Amount of MR recession				
Severe eyes	<0.001	3.893 (2.038 to 5.748)	<0.001	3.002 (1.558 to 4.446)
Mild eyes	0.002	5.742 (2.407 to 9.078)	-	-
Amount of superior MR transposition				
Severe eyes	0.093	62.079 (−11.452 to 135.611)		
Mild eyes	0.002	104.408 (43.755 to 165.061)	-	-
Amount of SR resection				
Severe eyes	0.959	−0.219 (−8.985 to 8.547)		
Mild eyes	0.959	−0.219 (−8.985 to 8.547)		
Amount of temporal SR transposition				
Severe eyes	-	-		
Mild eyes	0.959	−1.992 (−81.685 to 77.700)		
Amount of IR recession				
Severe eyes	0.382	−2.744 (−9.177 to 3.689)		
Mild eyes	0.382	3.660 (−4.917 to 12.238)		
Amount of nasal IR transposition				
Severe eyes	-	-		
Mild eyes	0.382	21.962 (−29.503 to 73.426)		

DON, dysthyroid optic neuropathy; LR, lateral rectus; MR, medial rectus; SR, superior rectus; IR, inferior rectus; CI, confidence interval.

## Data Availability

The data that support the findings of this study are available upon reasonable request.

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
