# Peer review of "Lateral Rectus Muscle Resection for New-Onset Esotropia Following Medial Orbital Wall Decompression in Thyroid Eye Disease"

_medicina, 2025, doi:10.3390/medicina61040559_

Round 1

Reviewer 1 Report

Comments and Suggestions for Authors

The topic of the paper is current in strabology and talks about the surgical treatment of esotropia after decompression of the orbit in a serious eye disease that occurs in a diseased thyroid gland. This is the quality of this paper, how to treat this postoperative condition.

In the table 1 it is necessary to correct Post-decompression Hertel exophthalmometric value (mm) (letter l)

Author Response

Point-by-Point Response to Comments from Reviewers

 We wish to thank the reviewers for allotting valuable time to help us improve our manuscript. We value all the comments that were given to us and hope that the changes we made will make the manuscript more acceptable to the reviewers.  

 Reviewer #1

The topic of the paper is current in strabology and talks about the surgical treatment of esotropia after decompression of the orbit in a serious eye disease that occurs in a diseased thyroid gland. This is the quality of this paper, how to treat this postoperative condition.

 In the table 1 it is necessary to correct Post-decompression Hertel exophthalmometric value (mm) (letter l)

 Reply:  Thank you for your kind comment. We have amended the manuscript to address this typographical error.

Reviewer 2 Report

Comments and Suggestions for Authors

This study evaluates the outcomes of lateral rectus (LR) muscle resection in patients with thyroid eye disease (TED) who developed esotropia following medial orbital wall decompression. It is a retrospective observational study involving 20 patients, assessing ocular deviation angles and binocular single vision (BSV) before and three months after surgery. The study reports a high surgical success rate (90%) and identifies key factors influencing the reduction of esodeviation angle, including dysthyroid optic neuropathy (DON), LR muscle resection in mild eyes, and medial rectus (MR) muscle recession in severe eyes. The authors conclude that LR muscle resection is effective in restoring ocular alignment in TED patients with post-decompression esotropia.

Major point:

  • The study provides a compelling justification for investigating LR muscle resection as an alternative to MR muscle recession for post-decompression esotropia. However, the introduction could better differentiate this study from prior research that included mixed patient groups (TED with and without decompression).
  • The criteria for surgical success (≤5° esodeviation and BSV grade B3 or better) are clearly defined, but additional justification for this cutoff is recommended. Are these thresholds commonly used in similar studies?
  • The statistical analysis section lacks details on potential confounders considered in multivariate analysis. Please provide a rationale for the chosen surgical success criteria and clarify how potential confounders (e.g., baseline TED severity, proptosis reduction) were controlled in the statistical models.
  • The study finds that combined LR muscle resection and MR muscle recession have less predictable outcomes in large-angle esotropia cases. This is an important finding, but the clinical implications are not sufficiently discussed.
  • The authors acknowledge limitations such as the short follow-up (three months) and retrospective design. However, additional discussion on potential long-term concerns (e.g., stability of ocular alignment, risk of late-onset overcorrection) would be valuable.
  • Figure 2 (patient face photos) could be supplemented with anonymized pre- and post-operative Hess chart images for more objective visualization. And the Hess chart methodology using ImageJ should specify whether inter-observer reliability was assessed.

Minor opinions:

  • Line 10: revise “were done one day” to “were performed one day”.
  • Line 14: revise “TED since it is” to “TED as it is”.
  • Line 22: revise “Six patients required additional” to “Six patients underwent additional”
  • Line 5: revise “New-onset or deteriorated esotropia post-decompression has been highly attributed to post-operative centrifugal displacement of a weakened MR muscle.” To “New-onset or worsening esotropia after decompression has been strongly linked to post-operative centrifugal displacement of a weakened MR muscle”.

Comments on the Quality of English Language

The English in this manuscript is generally clear and comprehensible, but minor grammatical errors, awkward phrasing, and structural inconsistencies are present. Some sentences could be reworded for better readability and clarity. Improving sentence flow and refining technical descriptions would enhance the overall quality of presentation. A careful language edit is recommended to ensure precision and fluency.

Author Response

Point-by-Point Response to Comments from Reviewers

 We wish to thank the reviewers for allotting valuable time to help us improve our manuscript. We value all the comments that were given to us and hope that the changes we made will make the manuscript more acceptable to the reviewers.  

Reviewer#2

Comment 1: The study provides a compelling justification for investigating LR muscle resection as an alternative to MR muscle recession for post-decompression esotropia. However, the introduction could better differentiate this study from prior research that included mixed patient groups (TED with and without decompression).

 Reply: We appreciate this valuable comment.  We have included results of studies with both decompression and non-decompression patients in our introduction. 

 Comment 2:  The criteria for surgical success (≤5° esodeviation and BSV grade B3 or better) are clearly defined, but additional justification for this cutoff is recommended. Are these thresholds commonly used in similar studies?

 Reply:  Thank you for your comment.  While no consensus has been made to clearly define the success rate of strabismus surgery, a horizontal deviation of ≤5° (vertical deviation of ≤2.5°) is generally acceptable for successful alignment with no diplopia. Some studies have used broader thresholds, such as ≤8° and ≤10° (https://doi.org/10.1016/j.jaapos.2012.10.019; https://doi.org/10.1186/s13005-024-00423-3),

 but we chose a stricter cutoff to ensure better functional and aesthetic outcomes. Similarly, the BSV grade of B3 or better includes the primary gaze, which is considered satisfactory for daily activities. We have cited additional studies that used similar parameters for defining success rates in the methodology section.

Comment 3:  The statistical analysis section lacks details on potential confounders considered in multivariate analysis. Please provide a rationale for the chosen surgical success criteria and clarify how potential confounders (e.g., baseline TED severity, proptosis reduction) were controlled in the statistical models.

 Reply:  Thank you for your valuable feedback. We analyzed multicollinearity among the variables that were correlated with the change in esodeviation angle, but there was no multicollinearity among these factors. We have amended our manuscript to better highlight this approach. Additionally, the rationale for the chosen surgical success criteria has been included.

Comment 4: The study finds that combined LR muscle resection and MR muscle recession have less predictable outcomes in large-angle esotropia cases. This is an important finding, but the clinical implications are not sufficiently discussed.

 Reply: Thank you for your thoughtful comment. We have now included a more detailed discussion of the clinical implications of this finding. We believe it is important for both surgeons and patients to be aware of the potential unpredictability of outcomes. Surgeons should inform patients about the possibility of needing additional interventions or adjustments, and may want to explore alternative surgical approaches to optimize outcomes. We hope this additional discussion provides clearer guidance on how these findings might influence clinical practice.

Comment 5:  The authors acknowledge limitations such as the short follow-up (three months) and retrospective design. However, additional discussion on potential long-term concerns (e.g., stability of ocular alignment, risk of late-onset overcorrection) would be valuable.

 Reply: We appreciate the suggestion to further discuss potential long-term concerns. In response, we have added previous studies that demonstrated favorable long-term results, with spontaneous resolution of late-onset overcorrections.

 Comment 6: Figure 2 (patient face photos) could be supplemented with anonymized pre- and post-operative Hess chart images for more objective visualization. And the Hess chart methodology using ImageJ should specify whether inter-observer reliability was assessed.

 Reply: We agree that including anonymized pre- and post-operative Hess chart images would provide a clearer representation of the outcomes. We have updated Figure 2 to include these images. Regarding the Hess chart methodology, the measurements were automated using ImageJ software, based on a previous study. Inter-observer reliability was not required. We have amended the methodology section to specify this.

Comment 7:  The English in this manuscript is generally clear and comprehensible, but minor grammatical errors, awkward phrasing, and structural inconsistencies are present. Some sentences could be reworded for better readability and clarity. Improving sentence flow and refining technical descriptions would enhance the overall quality of presentation. A careful language edit is recommended to ensure precision and fluency.  

Minor opinions: 

              Line 10: revise were done one day” to were performed one day”.

              Line 14: revise TED since it is” to TED as it is”.

              Line 22: revise Six patients required additional” to Six patients underwent additional

Line 5: revise New-onset or deteriorated esotropia post-decompression has been highly     attributed to post-operative centrifugal displacement of a weakened MR muscle.” ToNew- onset or worsening esotropia after decompression has been strongly linked to post-operative centrifugal displacement of a weakened MR muscle”.

Reply:  Thank you for your suggestions. We have incorporated these revisions and thoroughly proofread and edited the entire manuscript to improve readability.